# Peer review of "Single-Wall Torsion-Flux Realisation of Duality between M-Theory and Type~I String Theory"

_SciPost Physics_

## Round 1 · Referee Report · Anonymous (Referee 1) · 2025-12-22

Strengths

1- The manuscript attempts to address a significant and technically demanding topic: the direct geometric formulation of the duality between M-theory and Type I string theory.

2- The authors aim to synthesize a wide array of advanced theoretical machinery, including torsion fluxes, the Atiyah-Hirzebruch spectral sequence, and anomaly inflow, into a single framework.

Weaknesses

1- Fundamentally Flawed Geometry: The central premise of a "single-wall" background is topologically incoherent. The proposed diagonal involution on a torus possesses four fixed points, not one, and does not topologically identify the two distinct Hořava-Witten boundaries.

Furthermore, the geometric construction fails basic dimensional checks: the authors assign a rank-5 normal bundle to a codimension-1 hypersurface, which would imply an ambient spacetime of 15 dimensions rather than 11 . The authors never explain where the RP^4 they they say they want to discuss would arise from. (Usually in this context it arises as the normal sphere to an MO5 plane, but this is not what the authors consider.)

On top of that, throughout the text the M-theory spacetime is claimed to be the X^{9,1} x S^1 x S^1 which however is a 12-manifold that the authors refer to as an 11-manifold.

2- False Claims of Novelty: The manuscript frames the direct correspondence between M-theory and Type I string theory as a "hitherto unexplored" discovery. This contradicts the foundational literature; the original Hořava-Witten paper (cited by the authors) explicitly establishes that Type I duality follows from the symmetries of the 11-dimensional theory.

3- Mathematical Inconsistency: The text contains explicit category errors in algebraic topology. For example, the proposed differential (2.7) in the Atiyah-Hirzebruch spectral sequence attempts to sum an operator of degree 3 (Sq^3) with an operator of degree 4 (cup product with a 4-form), which is mathematically impossible. Similarly, the authors incorrectly identify the product of two 2-manifolds (RP^2 times Klein bottle) as a 3-cycle , which is not even a beginner's mistake.

In fact, throughout the text, the M-theory 4-form is conflated with the 3-form twist of K-theory.

Furthermore, the derivation of the flux quantization condition contains a direct logical contradiction. In the transition from Eq. (2.5) to (2.6), the authors explicitly state that the class λ vanishes (λ=0) due to stable triviality. However, they immediately proceed to claim that the "torsion reduction" of this zero class. Declaring a class to be zero while simultaneously using its non-zero reduction is mathematically absurd.

4- Erratic Exposition and Structural Amnesia: The logical flow of the manuscript is disjointed. Advanced tools like the Atiyah-Hirzebruch Spectral Sequence are claimed to imply results in Section 2 but are introduced in Section 6. Similarly, the 11D action functional is defined in Section 3 and then re-introduced in Section 5 as if it were a new concept, suggesting a lack of global coherence in the writing.

5- Misrepresentation of Standard Literature: The manuscript attributes a specific quantization formula to Witten (Ref. [7]) which does not exist in the cited text. It also claims that M-brane charges are classified by K-theory, citing Diaconescu, Moore, and Witten (Ref. [11]), whereas that reference discusses the derivation of Type IIA K-theory via dimensional reduction.

6- Reliability of Content: The density of basic errors—ranging from arithmetic failures (both 10+5 = 11 and 10+2 = 11 ) to spurious LaTeX artifacts in equations —suggests that the manuscript was not constructed with a genuine understanding of the physical or mathematical subject matter. These flaws are so encompassing that they render the evaluation of the paper's more detailed dynamical claims moot.

Report

This submission fails to meet the minimum standards for scientific publication and does not satisfy any of the SciPost acceptance criteria.

--Evaluation against General Acceptance Criteria:-- The manuscript fails the "General acceptance criteria" on multiple counts, specifically:

  1. concering the criterion"Free of... ambiguities and misrepresentations": The text is rife with fundamental misrepresentations.

  2. It attributes a formula to Witten (Ref. [7]) that does not exist in the cited source.

  3. It misrepresents the content of Diaconescu, Moore, and Witten (Ref. [11]) to claim that M-brane charges are classified by K-theory, whereas the reference discusses Type IIA K-theory via dimensional reduction.

  4. It contains spurious LaTeX artifacts in the equations (in (2.5) and (2.6)).

  5. concerning the criterion "Provide sufficient details... so that arguments... can be reproduced": The arguments are logically irreproducible because they are mathematically incoherent.

  6. The geometric setup is inconsistent: the authors define a "fixed ten-plane" but assign it a normal bundle of rank 5, which would imply a 15-dimensional spacetime, not 11-dimensional.

  7. Throughout the text, the authors define the ambient spacetime as a 12-dimensional manifold while referring to it as "11-dimensional M-theory."

  8. The derivation using the Atiyah-Hirzebruch Spectral Sequence is formally invalid, as it attempts to sum operators of different cohomological degrees (degree 3 and degree 4).

  9. concerning the criterion "Provide citations... in a way that is as representative and complete as possible": The citation list includes at least one dubious entry (Ref. [15], citing an "Authorea" preprint) and misinterprets standard literature (Hořava-Witten, Ref. [5]) to claim novelty for a result that was already discussed in the original paper.

--Evaluation against Expectations:-- The manuscript meets none of the required expectations for novelty or synergy:

  • The claimed "novel and synergetic link" between M-theory and Type I strings is physically ill-defined due to the geometric errors listed above.

  • Furthermore, the "direct" relation the authors claim to uncover is not new; it was explicitly discussed as a consequence of the symmetries of M-theory in the foundational Hořava-Witten paper itself.

--Conclusion:-- This manuscript appears to be a composite of theoretical physics jargon generated without underlying comprehension or mathematical verification. It contains fatal geometric contradictions, falsely attributed formulas, and smells of nonstandard text generation. It is scientifically unsound.

-Recommendation:-- I do not recommend publication in SciPost Physics and I would not recommend publication in any other scientific journal.

Requested changes

Although I am recommending rejection, if the authors were to attempt a revision, the following fundamental issues would need to be addressed:

1- Clarification of the Geometry: The authors must properly derive or motivate the RP^4 geometry which they claim they want to be discussing. In standard literature, such a geometry typically arises as the linking sphere of M5-branes probing orientifold planes. If this is the physical setup the authors intend to describe, they must state so clearly and explicitly distinguish their results from existing literature on M5-brane/O-plane dynamics. Currently, its appearance is unmotivated and context-free.

2- Correct Classification of M-brane Charges: The authors must correct their claim that M-brane charges are classified by standard twisted K-theory. If their intention is to discuss M-brane charges in the context of a generalized cohomology theory, they must engage with the actual literature on the subject—for instance, proposals involving quantization in ordinary shifted cohomology (as discussed by Witten et al. and Freed et al.), elliptic cohomology, or twisted Cohomotopy. They must remove the false attribution of K-theoretic M-brane classification to Diaconescu, Moore, and Witten (Ref. [11]).

Recommendation

Reject

---

## Editorial Decision

in_refereeing